# The Diagnostic Utility of Cell-Free DNA from Ex Vivo Bronchoalveolar Lavage Fluid in Lung Cancer

**DOI:** 10.3390/cancers14071764

**Published:** 2022-03-30

**Authors:** Sotaro Otake, Taichiro Goto, Rumi Higuchi, Takahiro Nakagomi, Yosuke Hirotsu, Kenji Amemiya, Toshio Oyama, Hitoshi Mochizuki, Masao Omata

**Affiliations:** 1Lung Cancer and Respiratory Disease Center, Yamanashi Central Hospital, Yamanashi 400-8506, Japan; ootake.sotaro.gx@mail.hosp.go.jp (S.O.); r-higuchi1504@ych.pref.yamanashi.jp (R.H.); nakagomi-chsz@ych.pref.yamanashi.jp (T.N.); 2Genome Analysis Center, Yamanashi Central Hospital, Yamanashi 400-8506, Japan; hirotsu-bdyu@ych.pref.yamanashi.jp (Y.H.); amemiya-bdcd@ych.pref.yamanashi.jp (K.A.); h-mochiduki2a@ych.pref.yamanashi.jp (H.M.); m-omata0901@ych.pref.yamanashi.jp (M.O.); 3Department of Pathology, Yamanashi Central Hospital, Yamanashi 400-8506, Japan; t-oyama@ych.pref.yamanashi.jp; 4Department of Gastroenterology, The University of Tokyo Hospital, Tokyo 113-8655, Japan

**Keywords:** lung cancer, cell-free DNA, airway, sequencing, mutation, genomic diagnosis

## Abstract

**Simple Summary:**

This study aims to detect cell-free DNA released from lung cancer cells into the airway using the ex vivo BAL model of our own establishing. We finally demonstrated that cell-free DNA released from lung cancer cells is more abundant in the airway than in the blood, and the efficient collection of cell-free DNA derived from lung cancer in the airway by BAL and its genomic analysis could allow the accurate diagnosis of lung cancer. We believe that this approach will possibly make a breakthrough in the currently unsatisfactory diagnostic yield for lung cancer, since it is a new and constitutive diagnostic focusing on the gene mutations of lung cancer and their release into the airway in the form of cell-free DNA.

**Abstract:**

Although bronchoscopy is generally performed to diagnose lung cancer, its diagnostic yield remains unsatisfactory. Assuming that lung cancer cells release cell-free DNA into the epithelial lining fluid, we hypothesized that lung cancer could be diagnosed by analyzing gene mutations in cell-free DNA in this fluid. This study included 32 patients with lung cancer who underwent surgery at our hospital. Bronchoalveolar lavage (BAL) was performed on the resected lung samples (ex vivo BAL model) after lobectomy. Each DNA sample (i.e., BAL fluid, primary lesion, and plasma) underwent deep targeted sequencing. Gene mutation analyses in the BAL fluid samples identified mutations identical to those in the primary lesions in 30 (93.8%) of 32 patients. In contrast, the microscopic cytology of the same BAL fluid samples yielded a diagnosis of lung cancer in only one of 32 patients, and the analysis of plasma samples revealed gene mutations identical to those in the primary lesions in only one of 32 patients. In conclusion, cell-free DNA released from lung cancer cells exists more abundantly in the airway than in the blood. The collection and analysis of the BAL fluid containing cell-free DNA derived from lung cancer can thus allow lung cancer diagnosis and the screening of driver mutations.

## 1. Introduction

Although bronchoscopy is generally performed to diagnose lung cancer, its diagnostic yield is unsatisfactory and is reported to be approximately 40% to 60% for cancers located in the peripheral lungs [1,2]. New technologies, including ultrathin fiberscope, navigation systems, endobronchial ultrasound, and new testing systems (ROSE: rapid onsite cytology) have been developed and improved in the last 20 years [3,4,5]. Although this has improved the diagnostic yield in recent years, further improvement is necessary.

At present, bronchoalveolar lavage (BAL) is performed to diagnose interstitial pneumonia and respiratory infection in general respiratory practice. However, BAL is not generally performed for diagnosing lung cancer because it has been occasionally reported to be associated with respiratory failure (0.46%), pneumonia (0.19%), and bronchial asthma (0.06%) [6]. In this study, we developed an ex vivo BAL model to explore and evaluate the possibility of lung cancer diagnosis using the BAL fluid samples and to simulate the actual clinical practice of BAL, thus allowing the diagnostic performance of BAL to be evaluated in the human lung without the concerns of complications.

To date, very few studies have examined the release of cell-free DNA (cfDNA) into the airway [7]. Regarding the biology of lung cancer, this study aimed to evaluate the possibility of cfDNA being released into the epithelial lining fluid (ELF) by comparing it with circulating tumor DNA being released into the peripheral blood. We also evaluated the pre-clinical model on the assumption that the additional genomic analysis of the BAL fluid might contribute to lung cancer diagnosis. Overall, this study presents basic data for the clinical application of this model to establish new clinical standards in “the BAL procedure for lung cancer diagnosis or gene mutation screening”.

## 2. Materials and Methods

### 2.1. Study Population

A total of 32 consecutive patients who underwent lobectomy for primary lung cancer at our hospital between April 2020 and October 2020 were enrolled in this study. Written informed consent was obtained from all the participants before the genetic research studies. This study was conducted in compliance with the ethical principles in the WMA Declaration of Helsinki and approved by the Institutional Review Board Committee of Yamanashi Central Hospital (protocol code Clin27-1). Medical data of the participants were collected from the cancer registry database of our institution, which included preoperative characteristics, computed tomography (CT) findings (tumor size and location), the operative procedure, and histopathological diagnosis, including lymph node metastasis status.

Cancers located in the outer, inner, and middle lung fields as primary lesions were referred to as peripheral, central, and middle lung cancers, respectively. Tumors were classified according to the World Health Organization’s classification (fifth edition) and clinical staging was based on the International Union Against Cancer tumor-node-metastasis classification (eighth edition) [8,9].

### 2.2. Ex Vivo BAL Model

BAL was performed on resected lung samples (ex vivo BAL model) placed on a back table immediately after lobectomy. A syringe tip was wedged into the bronchus with the segment harboring the tumor, 50 mL of physiological saline was injected a total of three times, and approximately 50 to 70 mL of the BAL fluid was collected (Figure 1). The recovery rate was 37.2% ± 9.6%. In addition to genomic analysis, the specimen aliquots were diagnosed cytologically by two specialist pulmonary pathologists.

### 2.3. Isolation and Purification of Genomic DNA Samples from Specimens

Approximately 10 mL of peripheral blood was drawn from the patients just prior to surgery, of which approximately 5–6 mL of plasma from each patient was used for analyses. The ex vivo BAL fluids were collected into sterile centrifuge tubes. After collecting the blood and BAL fluids, they were immediately sent to the genome analysis center of our hospital and centrifugation was initiated within 1 h of the acquisition of each sample. DNA preparation methods were essentially described in our previous report [7]. Briefly, the buffy coat (blood) and cell pellets (BAL fluid) were isolated after the centrifugation of these samples at 820× *g* for 10 min at 25 °C. Plasma from the blood and the supernatant of the BAL fluid was centrifuged at 20,000× *g* for 10 min and the resultant supernatant was subsequently transferred to sterile tubes and stored at −80 °C before DNA extraction**.** To obtain DNA from tumors, the formalin-fixed, paraffin-embedded (FFPE) tissue sections were first stained with hematoxylin and eosin, and the tumor lesions were microdissected using the ArcturusXT laser-capture microdissection system for the following DNA extraction as also previously described [7,10,11,12].

### 2.4. Quality Assessment of DNA Samples by Quantitative Real-Time PCR

The fragmentation of genomic DNA from FFPE tissues was validated using the TaqMan RNase P Detection Reagents Kit and FFPE DNA QC Assay on the 7th generation real-time PCR platform (ViiA™ 7 Real-Time PCR system, Thermo Fisher Scientific, Madison, WI, USA). DNA concentrations of sample DNAs and control DNA included in the kit were determined by serial dilution, and fragmentation levels were estimated by comparing the long versus short PCR products, as previously described in more detail [13,14].

### 2.5. Deep Sequencing for the Detection of Targeted Lung Cancer-Related Gene Mutations and Data Analysis

We have focused on 53 major genes that have been shown to be involved or implicated in lung cancer formation (Appendix A), which are described in The Cancer Genome Atlas [15,16]. Targeted deep sequencing was performed as described previously [17,18,19,20,21]. Briefly, primer design and library construction were carried out using Ion AmpliSeq™ Designer and Library Kit 2.0, respectively (Thermo Fisher Scientific). Sequence data were acquired using an ION Proton™ next-generation sequencing (NGS) system in conjunction with OneTouch™ 2 for template preparation and enrichment (Thermo Fisher Scientific) [7]. The data for tumor mutations (single nucleotide substitutions, insertions, and deletions) were analyzed using the ION Reporter™ Server system by comparing them with data of lymphocytes as a control [22,23,24,25] and visualized using the Integrative Genomics Viewer [26]. To avoid false-positive results, we used the following filtering parameters for variant calling: (1) the minimum number of variant allele reads was ≥10, (2) the coverage depth was ≥100, (3) variant allele fraction (VAF) ≥0.01, (4) UCSC Common SNPs = Not In, and (5) Confident Somatic Variants = In. High-confidence somatic mutations (i.e., >1% of total reads) were further searched in the bronchial wash fluid, sputum, and plasma specimens. The oncogenic relevance of single nucleotide and indel mutations was estimated using the Hidden Markov Models prediction in the COSMIC database [27] and the Precision Oncology Knowledge Base, OncoKB [28], respectively.

### 2.6. OncomineTarget Dx

The Oncomine Dx Target Test Multi-CDx system (Thermo Fisher Scientific) was developed to detect point mutations, short insertions and deletions (indels), gene rearrangements in cancer-related genes (e.g., BRAF, EGFR, *ROS1*, and *ALK*), allowing the following personalized chemotherapy with corresponding specific anti-tumor drugs [29,30]. This Oncomine system was also applied to the ex vivo BAL fluid specimens (10 ng DNA or RNA in total) according to the manufacturer’s instructions.

### 2.7. Statistical Analyses

The means with standard deviations (Mean ± SD) for continuous variables are listed in the Tables. Categorical variables were analyzed using the χ^2^ test to compare the groups. Correlation was determined (*p* < 0.05) with the use of Pearson’s correlation coefficient calculated with the JMP software (SAS Institute, Cary, NC, USA).

## 3. Results

### 3.1. Patient Characteristics

The 32 patients recruited in this study were categorized by their demographics (Table 1, Appendix A): age (range, 49–88 years; mean age, 70.0 ± 11.5 years); sex (12 males and 20 females); smoking history (19 smokers and 13 non-smokers); tumor location (24 peripheral, 1 central and 7 middle); histology (22 adenocarcinoma, 6 squamous cell carcinoma, 2 pleomorphic carcinoma, 1 adenosquamous cell carcinoma and 1 large cell neuroendocrine carcinoma); and pathological staging (stage I (*n* = 25), II (*n* = 3), III (*n* = 3), and IV (*n* = 1)). Maximum tumor sizes ranged from 13 to 54 mm (mean diameter, 24.3 ± 14.3 mm). Only 10 patients underwent bronchoscopy preoperatively; among them, four patients were diagnosed with lung cancer based on pathological findings.

### 3.2. Case Presentations

Presented below is the detailed patient information on two representative cases.

In case 1 (69-year-old woman), a tumor in right segment 1 was revealed by a CT scan (Figure 2A) and resected by right upper lobectomy. The tumor was diagnosed as a pathological stage IB invasive adenocarcinoma (Figure 2B).

The genomic analysis of the ex vivo BAL fluid and blood plasma identified mutations in the epidermal growth factor receptor (EGFR) (p.L858R) gene in the supernatant and cell fraction of the BAL fluid (Figure 2C, Appendix A), but not in blood plasma. When the Oncomine precision assay was performed on the ex vivo BAL fluid, the EGFR (p.Leu858Arg) mutation was detected.

In case 2 (87-year-old woman), similarly to case 1, a tumor in right segment 1 was found by CT screening (Figure 2D) and treated by a right upper lobectomy. The tumor was also diagnosed as a pathological stage IB invasive adenocarcinoma (Figure 2E). Mutations in the MET (p.D1028N) and KMT2D (p.E481K) genes were identified in her ex vivo BAL fluid, but not in her plasma, as shown (Figure 2F, Appendix A). When the Oncomine precision assay was performed on the ex vivo BAL fluid, the MET (p.D1028N) mutation was detected.

### 3.3. Targeted Sequencing Identified Shared Mutations in the Ex Vivo BAL Fluid and Lung Cancers

We employed targeted deep sequencing on the ex vivo BAL fluid samples of the 32 surgically resected tumors, with cognate blood cell samples as normal controls, from the 32 patients. The mean depths of sequencing coverage were 1534-fold for cancer samples (ranging from 143- to 6024-fold) and 1422-fold for normal controls (ranging from 105- to 7320-fold). A total of 92 somatic mutations (one to eight mutations per tumor) were identified by the targeted sequencing (detected mutant allele fractions ≥ 1%) from the 32 ex vivo BAL fluid samples (Appendix A).

When DNA with mutations homologous to those in the primary lung cancer lesions was detected in the BAL fluid, the detected DNA was identified as dispersed tumor DNA. The analysis of gene mutations in the BAL fluid samples identified gene mutations identical to those in the primary lesions in 30 (93.8%) of 32 patients, allowing the genomic diagnosis of lung cancer (Figure 3). Importantly, in these cases, the nucleotide position and mutation variance between the tumors and the BAL fluid were entirely consistent (Appendix A). When the analysis was restricted to oncogenic mutations, these were detected in 29 (90.6%) of 32 patients. This indicates that even in the preoperative stage when a comparison with the primary lesions was impossible, these 29 patients could be diagnosed as having lung cancer based on the presence of DNA with oncogenic mutations in the BAL fluid. The discordant mutations detected in either the FFPE sample or the BAL fluid are shown in Figure 4.

### 3.4. Analysis of the Mutational Overlap across Different Sample Types

In 32 patients, a total of 92 mutations were detected in FFPE samples, and 66.3% (61/92) of these mutations were shared by FFPE and BAL samples (Figure 5A). Meanwhile, of 68 mutations detected in BAL samples, 89.7% (61/68) were shared by two sample types. Only seven mutations were exclusively detected in BAL samples, which may suggest the presence of mutations derived from non-cancerous cells or undetectable mutations present in FFPE samples (Figure 4). When the analysis was restricted to oncogenic mutations, 45 mutations (97.8%) were shared by FFPE and BAL samples among the 46 mutations detected in BAL samples (Figure 5B).

The allele fraction of the BAL fluid dispersed tumor DNA (dtDNA) was significantly higher in smokers than in non-smokers and in patients without adenocarcinoma than in those with adenocarcinoma (Appendix A), suggesting that tumors showing poorer differentiation histologically tend to release more dtDNA fragments. However, the allele fraction was not significantly correlated with tumor location, tumor size, stage, or histological findings. On the contrary, as the allele fraction of mutations in tumors increased, that of the same mutations in the BAL fluid tended to increase as well (correlation, *p* < 0.05) (Figure 6).

### 3.5. Microscopic Pathological Examination vs. Genomic DNA Sequencing Approach for Bronchoscopic Specimens

Our genomic sequencing approach successfully detected DNA mutations, with similar mutation spectra to those in the corresponding primary lesions, in the BAL fluid of 30 patients (93.8%), whereas conventional cytological examination did so for only one patient (3.1%) (Case 22 in Table 2 and Appendix A). This patient showed cytological findings strongly suggestive of malignancy (class IV), whereas the remaining 31 patients had non-diagnostic cytological specimens: 29 patients with class I, and two patients with class II. The genomic sequencing analysis of plasma specimens marginally diagnosed one patient with lung cancer, with a diagnostic yield of merely 3.1% (Case 2 in Table 2). Based on these findings, the diagnostic yield for the genomic analysis of the BAL fluid was much better than that for the pathological examinations of the BAL fluid or the genomic analysis of plasma specimens (Table 2).

### 3.6. Comparison between dtDNA Positive and Negative Cases

In the two patients who were undiagnosed by the genomic analysis of BAL (Cases 5 and 12 in Table 2 and Appendix A), the invasive tumor size was as small as 7.0 ± 4.2 mm. Furthermore, two patients exhibited mostly ground-glass opacity on imaging findings and were confirmed as having predominantly adenocarcinoma in situ (lepidic pattern) based on their pathological examination (Figure 7).

## 4. Discussion

The diagnostic yield for lung cancer by conventional bronchoscopic examination remains low [2]; in fact, conventional diagnostic methods have not succeeded in making a diagnosis of approximately 40% of the patients at our hospital who underwent surgery for suspected lung cancer [7]. We, therefore, presumed that recently developed high-speed sequencing techniques could be applied for a more accurate diagnosis of lung cancer in conjunction with the appropriate sample preparation method. In this study, we developed a novel sample isolation method (i.e., the ex vivo BAL method) for pre-clinical diagnosis and demonstrated that the genomic sequencing of the ex vivo BAL fluid samples is a highly sensitive, accurate, and implementable diagnostic strategy for lung cancer.

In previous studies, a bronchial genomic classifier for lung cancer was developed based on the microarray gene expression profiles of normal-like epithelial cells surrounding the tumors in mainstem bronchi and shown to improve the diagnostic performance of bronchoscopy [31,32]. In contrast, we here employed deep sequencing, instead of quantitative gene expression analysis, to obtain mutation profiles, potentially highly specific and less affected by sample purity, by utilizing genomic DNA released from malignant cells.

We have previously reported that the analysis of gene mutations in the bronchial wash fluid allowed us to diagnose lung cancer in 10/18 patients (55.6%) [7]. As bronchial wash is generally performed immediately after bronchoscopic abrasion, it is plausible that tumor cells might have spread in the airway because of the effect of mechanical stimulation by tumor curettage in our previous study. In the present study, we did not apply any mechanical treatments, such as abrasion, and performed BAL on the lung samples immediately after lobectomy. The results of the present study suggest that mutant DNA is constantly being released from primary lesions into the ELF. Moreover, higher diagnostic accuracy was obtained when the BAL fluid was used (93.8%) than when the bronchial wash fluid was used (55.6%). The present study thus demonstrated that the efficient collection of ELF by BAL could be applied for the diagnosis of lung cancer in actual clinical practice.

The identification of genomic DNA mutations in the BAL fluid may provide a new, excellent strategy for the bronchoscopic diagnosis of lung cancer. In our routine pathological examination setting, merely one (3.1%) out of 32 patient specimens was diagnosed as lung cancer, whereas the diagnosis rate by our genomic sequencing approach reached 93.8% (30 out of 32 patients). This considerable difference in detection yield is very likely due to fundamental differences in the sample isolation method and the detection strategy. This is partly because classification into cytological stages (I–V) by pathologists has often relied on their experience and subjective judgment, and thereby could have been variable. In contrast, genomic sequence data are highly objective digital information, and the results are thus more precise. The BAL fluid has been previously regarded as a material with limited utility value for diagnostic detection; however, we here demonstrated that this can be utilized for precision diagnosis by virtue of the “deep sequencing” platform.

Although cell-free DNA in peripheral blood (circulating tumor DNA (ctDNA)) has attracted attention as a new biomarker for lung cancer, there is limited scientific evidence to support its utility for detecting early-stage cancers. In our previous study, ctDNA was preoperatively detected in peripheral blood in five (16.7%) of 30 patients with lung cancer who underwent surgery [33]. The contribution of ctDNA to early diagnosis is thus limited. Ohira et al., also reported that the detection rates of ctDNA are 0% in stage IA and IB lung cancer and 16.7% in stage IIA and higher lung cancers, suggesting that tumor volume is a determinant of the feasibility of mutation detection with ctDNA [34]. Furthermore, since cfDNA are similarly present in healthy individuals, it is practically impossible to distinguish ctDNA from normal tissue-derived cfDNA. In addition, mutations of cfDNA from non-malignant tissues, especially from hematopoietic stem cells [35], have been reported. Specifically, the age-related acquisition of somatic mutations, referred to as the clonal hematopoiesis of indeterminate potential (CHIP), leading to clonal expansion in regenerating hematopoietic stem cell populations, is commonly detected in older individuals [36]. These clinical aspects of ctDNA from lung cancer are associated with many issues of sensitivity and specificity [37]. Thus, ctDNA remains far from being clinically applied. In this context, based on the fact that lung cancer derives from the epithelium, we presumed that DNA residing in the airway could be a diagnostic material with higher abundance and improved detectability compared to plasma ctDNA. This prompted us to test the BAL fluid for the detection of cancer-specific mutations in the pursuit of application to the clinical diagnosis of lung cancer. In fact, genomic DNA mutations, highly related to those identified in primary lesions, were detected in the ex vivo BAL fluid with a high rate (30/32 patients), which overwhelmingly outperformed the rate with the peripheral plasma (1/32 patients). These findings suggest that the ex vivo BAL fluid is clearly superior to peripheral blood as a specimen for DNA diagnosis.

In this study, NGS was performed for the purpose of comprehensively searching for gene mutations in cell-free DNA analysis in plasma and BAL fluid. NGS was performed on both samples with almost the same procedures, and we showed higher detection sensitivity for cell-free DNA in BAL fluid than ctDNA in plasma under the same methodology. However, since blood is a flowing fluid by nature, it may be appropriate to use a more precise assay to detect ctDNA in plasma. In fact, many studies have been reported on methods for the highly sensitive detection of the ctDNA of a specific gene by using droplet digital PCR [38,39,40]. On the other hand, according to the American Society of Clinical Oncology/College of American Pathologists, the mutation profile of peripheral blood cfDNA does not always coincide with that of tumor DNA of the same individual [41]. Cell-free DNA analysis can be performed on BAL and plasma samples using highly sensitive detection methods such as cancer personalized profiling by deep sequencing (CAPP-seq) and circulating single-molecule amplification and resequencing technology (cSMART) assay to directly compare the sensitivity and specificity of lung cancer diagnosis between the two samples [42,43,44,45], which we consider to be an important subject to be examined following the current study.

The seven mutations detected only in BAL and not in tumor tissue were not detected in blood, and we think that the possibility of CHIP is low. On the other hand, it has previously been reported that mutations accumulate on several genes in airway epithelial cells with smoking [46,47]. In our study, all four patients with these seven mutations were also smokers. We have speculated that the mutations detected this time may be attributed to the release/secretion of mutant DNA accumulated in airway epithelial cells and alveolar macrophages, in line with recent studies reporting the presence of cancer gene mutations in different noncancer tissues [48,49,50]. In fact, in the report of Roncarati et al., many mutant genes such as *KRAS*, *TP53*, and *PIK3CA* were detected in the bronchial wash fluid of cancer-free individuals [51]. Regarding the origin of these seven mutations, there is also a possibility that these mutations happened to not be detected at the tumor tissue analysis site due to intratumor heterogeneity, or that the results were a false positive due to sequencing errors [52]. In addition, Ryu et al., reported the NGS analysis results of bronchial wash fluid obtained from 12 lung cancer patients, showing that the mutant allele fraction in bronchial washing fluid was approximately 4% on average [53]. The mutant allele fraction in our BAL samples was an average of 11.2% and a median of 6%, and it can be said that the values were relatively high. However, considering the noncancerous cell-free DNA released into the epithelial lining fluid from a large number of alveolar macrophages and airway epithelial cells, the analysis results, showing that the mutant DNA in BAL fluid exhibits such a low value, can be considered to be reasonable, and these low values may also pose some challenges for the diagnosis of lung cancer.

Among the patients included in this study, lung cancer was diagnosed by comparing the mutation profiles of the primary lesion and the BAL fluid samples after surgery. In actual clinical practice, only the BAL fluid samples are available before surgery. An important issue is how diagnostic accuracy can be improved in this circumstance. Although the mutations detected in the airway might have included those derived from normal cells, similar to the case of mutations detected in blood, 45 (73.8%) of all 61 shared mutations detected in the tumor tissue and BAL fluid were oncogenic. Therefore, if the presence of oncogenic mutations were regarded as the presence of lung cancer, the examination of the BAL fluid samples alone on this basis could yield a diagnosis of lung cancer in 29 patients (90.6%) with oncogenic mutations present among all 32 patients. The application of this approach, including BAL, followed by deep sequencing, and then the screening of detected SNVs and indels for oncogenicity, can be expected to allow lung cancer diagnosis efficiently. As an advantage compared with conventional bronchoscopic biopsy, where lung cancer cannot be diagnosed unless the biopsy forceps are guided to the vicinity of the tumor under X-ray fluoroscopy and the tip of the forceps finally hits the lung tumor directly, lung cancer can be easily diagnosed by routine procedures because the identification of the segmental bronchus is sufficient for the BAL. In fact, the correct diagnosis rate of lung cancer by bronchoscopic biopsy is around 30–80%, depending on the size and location of the tumor [7,31], whereas if the combination of BAL procedure and genomic analysis can be realized, the diagnosis rate of lung cancer is expected to be significantly improved, regardless of the size or location of the tumor. On the other hand, disadvantages include prolonged examination time for bronchoscopy, increased invasion in patients, and the development of complications associated with BAL (many of which are minor). Although there seem few serious disadvantages regarding the BAL procedure, the study should be continued toward clinical application while paying careful attention to ethical aspects. The above-mentioned Ryu et al., performed bronchial washing in vivo in 12 patients with early-stage lung cancer and performed NGS analysis with resultant fluids [53]. As a result, they reported encouraging results showing that the concordance of driver mutations between bronchial washing fluids and primary tumors was 95.0% [53]. This suggests that this type of diagnostic approach is feasible and promising, thus possibly opening a new avenue for “precision medicine” in lung cancer diagnosis.

There are several limitations to this study. First, the physiological mechanisms and efficiency of DNA release from tumors into the airway remain to be elucidated. Relatively many mutant DNAs were detected only in tumors but not in BAL fluid, in a similar way as, upon the measurement of blood ctDNA, some but not all of the mutations harbored by the tumor are often detected in plasma [20,33,54]. Of the tumor mutations, it is still unclear which mutant DNA is released into the airway and bloodstream, by what timing, and by what mechanism, and this biological enigma is a topic for future study; secondly, the sample sizes may not be sufficiently large. The patients without lung cancer were not examined and the detection thresholds were not determined in this study.; thirdly, the ex vivo BAL model uses surgically resected lungs, and that is not exactly the same as the BAL performed in clinical practice. In the case of surgically removed lungs, BAL may be contaminated with blood and bronchial epithelia during intraoperative manipulation, and under the situation where there is no blood flow, BAL recovery efficiency and cell-free DNA release from the tumor may be affected. Therefore, future studies should validate whether the detection of dispersed mutant DNA is applicable to a broad range of lung cancer diagnoses toward the following personalized therapies. Nevertheless, our results clearly indicate that next-generation sequencing is a powerful and integral component of “precision medicine”. Besides the simple diagnosis of lung cancer, this detection approach has several diagnostic strengths: the detection of residual lesions, the monitoring of tumor changes over time, and the surveillance of the emergence of drug-resistant mutations. Since our major aim of this provisional study was to urge the detection approach for clinical development, this study with a medium sample size should provide crucial information and insights.

## 5. Conclusions

As cfDNA released from lung cancer cells is more abundant in the airway than in blood, the efficient collection of cfDNA derived from lung cancer in the airway by BAL and its genomic analysis could allow the accurate diagnosis of lung cancer and the screening of driver mutations. In this study, we have developed a new diagnostic approach of BAL followed by the analysis of gene mutations for lung cancer diagnosis. The application of this approach in clinical settings can facilitate an accurate diagnosis in most patients with lung cancer.

## Figures and Tables

**Figure 1 cancers-14-01764-f001:**
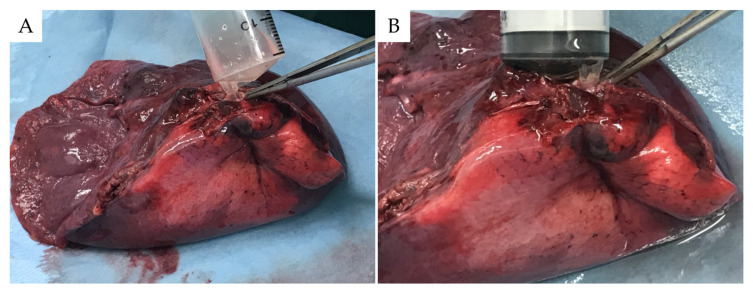
Ex vivo bronchoalveolar lavage (BAL) model. Immediately after lobectomy, a 50 mL syringe was wedged into the bronchus in the segment where a tumor existed, in the lung placed on the back table (**A**). BAL was performed using the same procedure performed in bronchoscopy (**B**).

**Figure 2 cancers-14-01764-f002:**
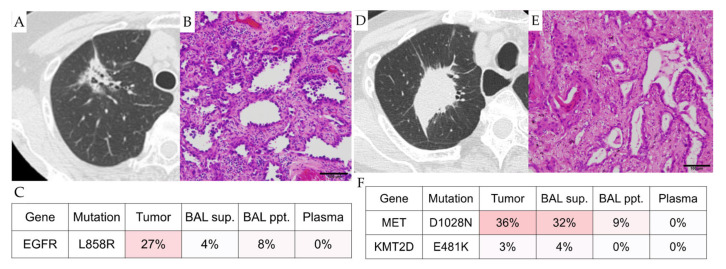
Representative Cases. (**A**–**C**): Case 1. (**A**), CT image; (**B**), histology; (**C**), heatmap of mutations detected in the samples. (**D**–**F**): Case 2. (**D**), CT image; (**E**), histology; (**F**), heatmap of mutations detected in the samples. The scale bars indicate 100 μm.

**Figure 3 cancers-14-01764-f003:**
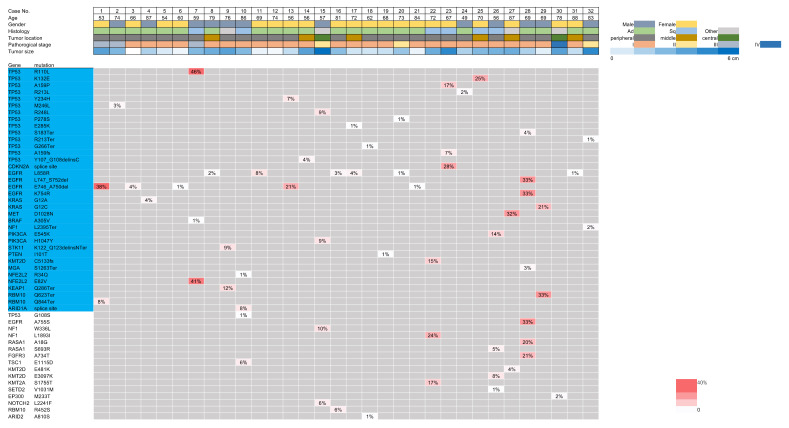
Shared mutations in the ex vivo bronchoalveolar lavage (BAL) fluid and lung cancer tissues. The shared mutations are shown with the allele fractions visualized on the heatmap. The gene mutations highlighted in light blue are oncogenic.

**Figure 4 cancers-14-01764-f004:**
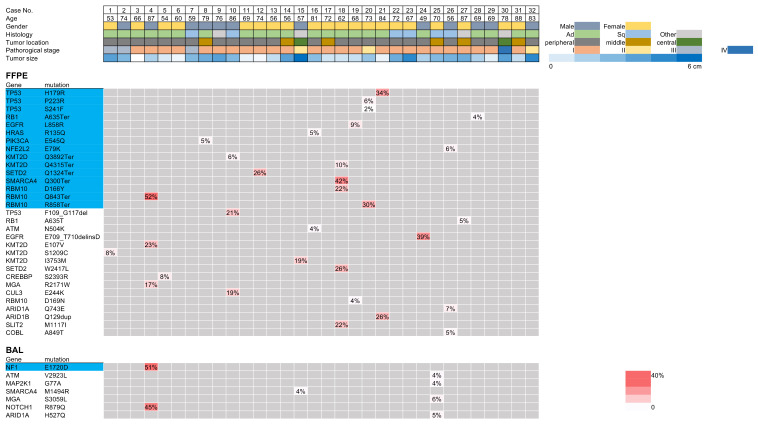
Discordant mutations between the ex vivo bronchoalveolar lavage (BAL) fluid and lung cancer tissues. The discordant mutations are shown with the allele fractions visualized on the heatmap. The upper rows show mutations detected only in the FFPE sample and the lower rows show mutations detected only in the BAL fluid. The gene mutations highlighted in light blue are oncogenic.

**Figure 5 cancers-14-01764-f005:**
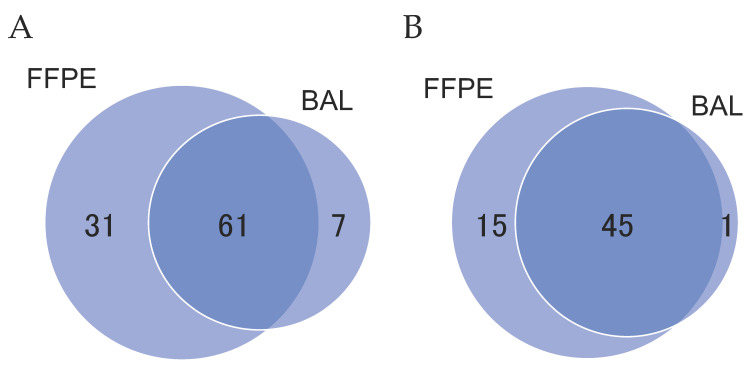
Venn diagram indicating the mutational overlap between formalin-fixed paraffin-embedded (FFPE) and bronchoalveolar lavage (BAL) samples. (**A**) All mutations; (**B**) oncogenic mutations.

**Figure 6 cancers-14-01764-f006:**
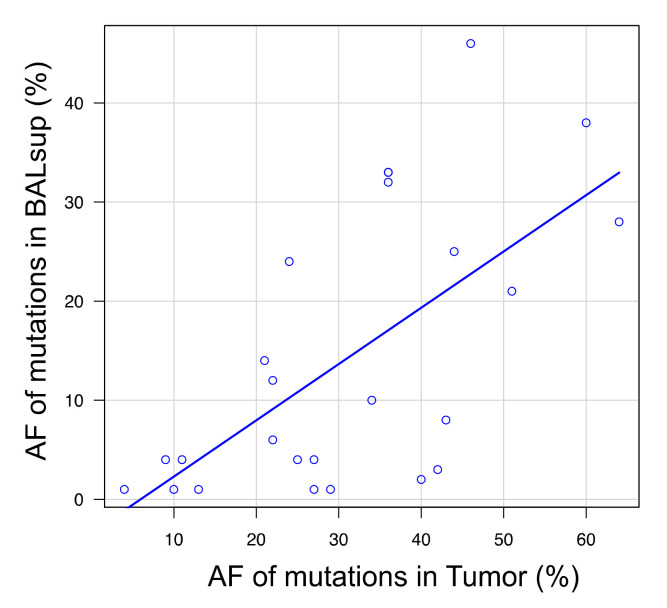
Correlation of allele fraction between mutations detected in the bronchoalveolar lavage (BAL) fluid and tumor.

**Figure 7 cancers-14-01764-f007:**
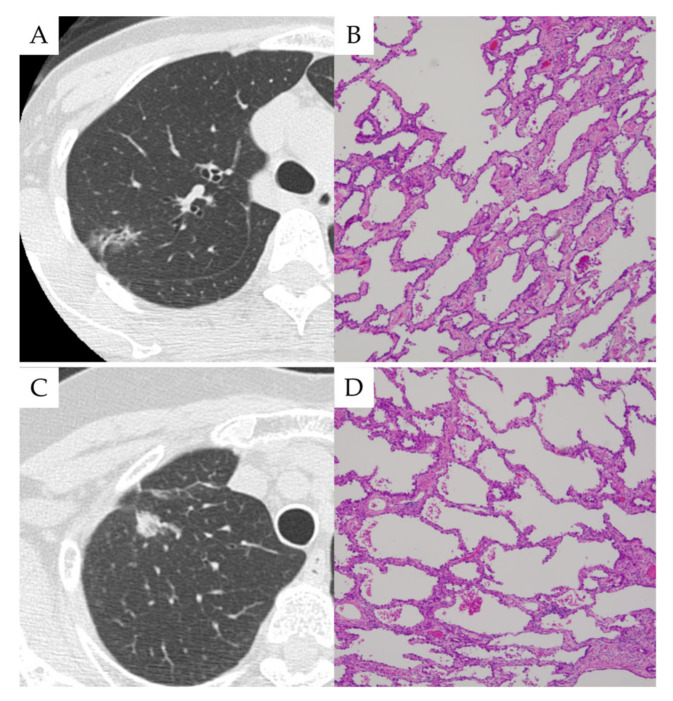
Two cases that showed no dispersed tumor DNA detection. (**A**,**B**) CT image and histology of case 1. (**C**,**D**) CT image and histology of case 2.

**Table 1 cancers-14-01764-t001:** Characteristics of the patients.

Parameters	Variables	Number
Total number		32
Age, years	Mean ± SD	70.0 ± 11.5
Sex	Male	12 (37.5%)
	Female	20 (62.5%)
Smoking history	Smoker	19 (59.4%)
	Non-smoker	13 (40.6%)
Tumor size, mm	Mean ± SD	24.3 ± 14.3
Tumor location	Central	1 (3.1%)
	Middle	7 (21.9%)
	Peripheral	24 (75%)
Histology	Adenocarcinoma	22 (68.8%)
	Squamous cell carcinoma	6 (18.8%)
	Pleomorphic carcinoma	2 (6.3%)
	Adenosquamous cell carcinoma	1 (3.1%)
	Large cell neuroendocrine carcinoma	1 (3.1%)
Pathological stage	I	25 (78.1%)
	II	3 (9.4%)
	III	3 (9.4%)
	IV	1 (3.1%)

**Table 2 cancers-14-01764-t002:** Conventional and genomic bronchoscopic diagnoses of lung cancer.

Case	1	2	3	4	5	6	7	8	9	10	11	12	13	14	15	16
Microscopic Dx																
Cytology	−	−	−	−	−	−	−	−	−	−	−	−	−	−	−	−
Genomic Dx																
BALF sup.	+	+	−	+	−	−	+	+	+	+	+	−	+	+	+	+
BALF ppt.	+	−	+	−	−	+	−	+	+	−	+	−	−	−	−	+
Plasma	−	+	−	−	−	−	−	−	−	−	−	−	−	−	−	−
Case	17	18	19	20	21	22	23	24	25	26	27	28	29	30	31	32
Microscopic Dx																
Cytology	−	−	−	−	−	+	−	−	−	−	−	−	−	−	−	−
Genomic Dx																
BALF sup.	+	+	+	+	−	+	+	+	+	+	+	+	+	−	+	−
BALF ppt.	−	−	−	−	+	+	+	−	−	−	+	+	−	+	−	+
Plasma	−	−	−	−	−	−	−	−	−	−	−	−	−	−	−	−

BALF, bronchoalveolar lavage fluid; sup, supernatant; ppt, precipitant; +, diagnosable; −, non-diagnosable.

## Data Availability

The data presented in this study are available on request from the corresponding author. The data are not publicly available due to them containing information that could compromise research participant privacy/consent.

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
