# Peer review of "The Diagnostic Utility of Cell-Free DNA from Ex Vivo Bronchoalveolar Lavage Fluid in Lung Cancer"

_cancers, 2022, doi:10.3390/cancers14071764_

Round 1

Reviewer 1 Report

In this study, Otake et al. showed that ex-vivo bronchoalveolar lavage fluid could be used to detect oncogenic mutations in early stages NSCLC. This study is of interest in the field to improve the role of BAL in the diagnosis of NSCLC especially for early stages. Despite the few number of sample included in this study, the results on BAL samples are interesting for application on daily practice.

Major point:

My major concern is about ctDNA analysis on plasma samples. In the discussion section, the authors indicated that the detection rates of ctDNA in stage I NSCLC remains extremely low to support their results. However, different studies reported fraction of patients with detectable preoperative ctDNA higher than 40% even in early stages NSCLC especially with NGS analysis (e.g. Peng M et al. Front Oncol. 2020;10:561598.doi: 10.3389/fonc.2020.561598 / Fernandez-Cuesta L et al. ebiomedicine 2016;10:117-23 / …). That’s why, the very low positivity rate of ctDNA seems really unexpected (i.e. 1 of 32 patients) and not fully convincing at this time.

Could the authors give details about the methodological approach of ctDNA? This method is too briefly described and the results not fully convincing at this time. This part of the manuscript should be improved.

- ctDNA samples collected on EDTA collecting tubes needs to be quickly centrifuged (within 2 hours) and stored at -80°C to avoid false negative results. Could this point be responsible of the low detection rate of ctDNA? Specific ctDNA collecting tubes are available to avoid preanalytical problems. Why the authors did not use these collecting tubes?

- The volume of extracted plasma is not indicated in the manuscript and should be added in the text. It is well established that this volume has a high impact on ctDNA detection rate

- Do the authors confirmed the negative result of ctDNA using a more sensitive approach such as ddPCR for recurrent oncogenic drivers (e.g. L858R, del19, ...)?

In this context, the low diagnosis accuracy of ctDNA vs BAL should be interpreted with caution. This point should be clarifying in the different sections of the manuscript.

Other points:

- 7 mutations (1 oncogenic mutation) were exclusively reported in BAL samples. Authors suggested that these mutations derived from non-cancerous cells. Do they think it could be clonal hematopoiesis? In this case, are these mutations also reported in plasma analysis? Are BAL non-cancerous cells mutations previously reported in the literature?

- Figure 3 shows different allele fractions close to 1%. 1% is frequently below the lower limit of detection of variants by NGS analysis. In these cases, it may be difficult to diagnose NSCLC using BAL sample without a comparison with tissue analysis to avoid false positive results. Could the authors clarify this point? It should be added in the discussion section.

- Table 3 compare 30 samples dtDNA+ vs only 2 samples (dtDNA-). Is the statistical analysis possible in this context? How could these results interpreted?

- The representation of the Venn diagram (figure 4) must be improved: the circles are not available in the reviewed version of the manuscript.

Author Response

Major point:

Could the authors give details about the methodological approach of ctDNA? This method is too briefly described and the results not fully convincing at this time. This part of the manuscript should be improved.

- ctDNA samples collected on EDTA collecting tubes needs to be quickly centrifuged (within 2 hours) and stored at -80°C to avoid false negative results. Could this point be responsible of the low detection rate of ctDNA? Specific ctDNA collecting tubes are available to avoid preanalytical problems. Why the authors did not use these collecting tubes?

Response: After collecting the blood and BAL fluids, they were immediately sent to the genome analysis center of our hospital. Centrifugation was initiated within 1 h of acquisition of each sample. The buffy coat (blood) and cell pellets (BAL fluid) were isolated after centrifugation of these samples at 820 ×g for 10 min at 25°C. Plasma from the blood and supernatant of the BAL fluid was centrifuged at 20,000 ×g for 10 min and the resultant supernatant was subsequently transferred to sterile tubes and stored at −80°C before DNA extraction.

Blood samples were collected in EDTA-2Na tubes, because we have continued to use these tubes in our research on ctDNA since 2014 and have had no problems with the detection and analysis of ctDNA (doi: 10.1007/s12032-016-0744-x. doi: 10.18632/oncotarget.19538. doi: 10.1016/j.ejca.2017.09.004.). However, since you have provided us the information that the cell-free DNA collection tube is commercially available and is effective for the storage of ctDNA, we would like to use these tubes for future study. Thank you for providing us the valuable information.

- The volume of extracted plasma is not indicated in the manuscript and should be added in the text. It is well established that this volume has a high impact on ctDNA detection rate

Response: Approximately 10 ml of peripheral blood was drawn from the patients just prior to surgery, of which approximately 5-6 ml of plasma samples from each patient was used for analyses. We added this description to the Methods section.

- Do the authors confirmed the negative result of ctDNA using a more sensitive approach such as ddPCR for recurrent oncogenic drivers (e.g. L858R, del19, ...)?

Response: Since the main purpose of this study was the comprehensive analysis of cell-free DNA in BAL, no more precise analysis was further performed for ctDNA in plasma.

In this context, the low diagnosis accuracy of ctDNA vs BAL should be interpreted with caution. This point should be clarifying in the different sections of the manuscript.

Response:

In this study, NGS was performed for the purpose of comprehensively searching for gene mutations in cell-free DNA analysis in plasma and BAL fluid. NGS was performed on both samples with almost the same procedures, and we showed higher detection sensitivity for cell-free DNA in BAL fluid than ctDNA in plasma under the same methodology. However, since blood is a flowing fluid by nature, it may be appropriate to use a more precise assay to detect ctDNA in plasma. In fact, as the reviewer suggested, many studies have been reported on methods for highly sensitive detection of ctDNA of a specific gene by using droplet digital PCR. On the other hand, according to the American Society of Clinical Oncology/College of American Pathologists, the mutation profile of peripheral blood cfDNA does not always coincide with that of tumor DNA of the same individual. Cell-free DNA analysis can be performed on BAL and plasma samples using highly sensitive detection methods such as CAPP-seq and cSMART assay to directly compare the sensitivity and specificity of lung cancer diagnosis between the two samples. We consider this to be an important subject to be examined following the current study.

We added these descriptions in the Discussion section.

Other points:

- 7 mutations (1 oncogenic mutation) were exclusively reported in BAL samples. Authors suggested that these mutations derived from non-cancerous cells. Do they think it could be clonal hematopoiesis? In this case, are these mutations also reported in plasma analysis? Are BAL non-cancerous cells mutations previously reported in the literature?

Response:

The 7 mutations detected only in BAL and not in tumor tissue were not detected in blood, and we think that the possibility of clonal hematopoiesis of indeterminate potential (CHIP) is low. On the other hand, it has previously been reported that mutations accumulate on several genes in airway epithelial cells with smoking (doi: 10.3390/cells10051173, doi: 10.1038/s41586-020-1961-1.). In this study, all 4 patients with these 7 mutations were also smokers. We have speculated that the mutations detected this time may be attributed to the release/secretion of mutant DNA accumulated in airway epithelial cells and alveolar macrophages, in line with recent studies reporting the presence of cancer-gene mutations in different noncancer tissues. In fact, in the report of Roncarati et al., many mutant genes such as KRAS, TP53, and PIK3CA were detected in the bronchial wash fluid of cancer-free individuals.

Regarding the origin of these 7 mutations, there is also a possibility that these mutations happened to be not detected at the tumor tissue analysis site due to intratumor heterogeneity or that the results were false positive due to sequencing errors.

- Figure 3 shows different allele fractions close to 1%. 1% is frequently below the lower limit of detection of variants by NGS analysis. In these cases, it may be difficult to diagnose NSCLC using BAL sample without a comparison with tissue analysis to avoid false positive results. Could the authors clarify this point? It should be added in the discussion section.

Response:

To avoid false positive results, we used the following filtering parameters for variant calling: (1) the minimum number of variant allele reads was ≥10, (2) the coverage depth was ≥ 100, (3) variant allele fraction (VAF) ≥ 0.01, (4) UCSC Common SNPs = Not In, and (5) Confident Somatic Variants = In.

Ryu et al. reported the NGS analysis results of bronchial wash fluid obtained from 12 lung cancer patients, showing that the mutant allele fraction in bronchial washing fluid was approximately 4% on average. The mutant allele fraction in our BAL samples was an average of 11.2% and a median of 6%, and it can be said that the values were relatively high. However, considering the noncancerous cell-free DNA released into the epithelial lining fluid from a large number of macrophages and airway epithelial cells present in the airway, the analysis results that the mutant DNA in BAL fluid exhibits such a low value can be considered to be reasonable, although these low values may pose some challenges for the diagnosis of lung cancer.

As the limitation of this study, patients without lung cancer were not examined and the detection thresholds were not determined.

We added these descriptions to the Methods and Discussion sections.

- Table 3 compare 30 samples dtDNA+ vs only 2 samples (dtDNA-). Is the statistical analysis possible in this context? How could these results interpreted?

Response: We agree with the reviewer that this analysis is statistically possible, but hard to interpret. Therefore, we deleted this table from the manuscript.

- The representation of the Venn diagram (figure 4) must be improved: the circles are not available in the reviewed version of the manuscript.

Response: Thank you for bringing it to our attention. All the reviewers pointed out the deformity of this figure. We think it may be due to the computer glitch, probably caused by the version of Word software we used. We will ask the editor to solve this problem.

Thank you for your thoughtful comments.

Reviewer 2 Report

The manuscript by Otake et al examines the utility of BAL targeted sequencing for diagnosis of lung cancer. Given that accurate diagnosis of lung cancers has been challenging, the proposed study is of high clinical significance.  Authors provide data demonstrating detection of tumor DNA mutations in 30 of 32 BAL samples. These results are encouraging and warrant further investigations in diagnostic utility of BAL sequencing in clinical settings. However, authors need to address the following concerns:

  • In addition to two false negatives (5 & 12), in many BAL samples only 1 oncogene in the panel was detected and in 5 BAL samples (6, 18, 19, 21, and 31) the 1-gene shared mutation was at 1%. Given the possibility of mutations in non and pre neoplastic lung due to smoking and the high burden of false positives in patient care, this level of mutation would perhaps not be adequate for diagnosis of cancer in clinical setting. Along the same lines, it is important to note that 7 mutations including in 1 oncogene was detected in BAL but not in the tumor. A further explanation of potential cancer detection thresholds in clinical settings is needed.
  • Regarding sensitivity and according to the Venn diagram (Figure 4), 1/3-1/4 of the mutations in tumors were not present in BAL, even though the BAL collections had large volumes in this experimental setting (50-70 ml). In authors view, is this volume of samples feasible in clinical care and how would the change in volume effect sensitivity?
  • Advantage and disadvantage of BAL sequencing in comparison with biopsies need to be discussed.
  • Prior art such as a report by Ryu (Pubmed ID: 31036768) for diagnosis of early-stage NSCLC have not been discussed.
  • Figure 3 shows only the shared mutations between BAL and tumor. It is important to also show the discrepant mutations between these two sample types for each patient either in the same or a separate figure.
  • The detection sensitivity in blood samples appear lower than prior reports. How much was the starting volume? Could that explain lower than expected sensitivity?
  • There seems to be discrepancies in tumor sizes between Tables 3 and S2. For example, the tumor sizes for patient 5 and 12 in Table S2 are 2.3 and 3.1 cm, respectively. But, in Table S2 those are listed as 0.7 cm.

Author Response

  • In addition to two false negatives (5 & 12), in many BAL samples only 1 oncogene in the panel was detected and in 5 BAL samples (6, 18, 19, 21, and 31) the 1-gene shared mutation was at 1%. Given the possibility of mutations in non and pre neoplastic lung due to smoking and the high burden of false positives in patient care, this level of mutation would perhaps not be adequate for diagnosis of cancer in clinical setting. Along the same lines, it is important to note that 7 mutations including in 1 oncogene was detected in BAL but not in the tumor. A further explanation of potential cancer detection thresholds in clinical settings is needed.

Response:

There are many unclear points regarding the mechanisms by which mutant DNA is released from the tumor into the airways, and upon measurement of blood ctDNA, some but not all of the mutations possessed by the tumor are often detected in plasma (doi: 10.1007/s12032-016-0744-x. doi: 10.18632/oncotarget.19538. doi: 10.1016/j.ejca.2017.09.004.).

In addition, Ryu et al. reported the NGS analysis results of bronchial wash fluid obtained from 12 lung cancer patients, showing that the mutant allele fraction in bronchial washing fluid was approximately 4% on average. The mutant allele fraction in our BAL samples was an average of 11.2% and a median of 6%, and it can be said that the values were relatively high. However, considering the noncancerous cell-free DNA released into the epithelial lining fluid from a large number of macrophages and airway epithelial cells present in the airway, the analysis results that the mutant DNA in BAL fluid exhibits such a low value can be considered to be reasonable, although these low values may pose some challenges for the diagnosis of lung cancer. As the limitation of this study, we mentioned that patients without lung cancer were not enrolled and the detection thresholds were not determined.

All 4 patients with 7 mutations detected only in BAL and not in tumor tissue were smokers. We have speculated that the mutations detected this time may be attributed to the release/secretion of mutant DNA accumulated in airway epithelial cells and alveolar macrophages, in line with recent studies reporting the presence of cancer-gene mutations in different noncancer tissues. In fact, in the report of Roncarati et al., many mutant genes such as KRAS, TP53, and PIK3CA were detected in the bronchial wash fluid of cancer-free individuals. In the future, we would like to continue studies using ex-vivo BAL, including non-lung cancer patients with a history of smoking, set detection thresholds useful for lung cancer diagnosis, and establish a diagnostic method.

We added these descriptions in the Discussion section.

  • Regarding sensitivity and according to the Venn diagram (Figure 4), 1/3-1/4 of the mutations in tumors were not present in BAL, even though the BAL collections had large volumes in this experimental setting (50-70 ml). In authors view, is this volume of samples feasible in clinical care and how would the change in volume effect sensitivity?

Response:

As the reviewer suggested, relatively many mutant DNAs were detected only in tumors but not in BAL. Of the tumor mutations, it is still unclear which mutant DNA is released into the airway and blood stream by what timing and by what mechanism, and this is a topic for future study. We added this description in the Discussion section.

Injecting 50 ml into the segmental bronchus three times was similar to the method used in usual medical care, and the recovery amount in our model was alao similar to that in usual medical care. The allelic fraction can be defined as the number of times a mutated base is observed, divided by the total number of times any base is observed at the locus. We only use the method of “50 ml x 3 times” to efficiently lavage the airway, and we do not think that this injection volume is a factor that greatly affects AF measurement.

  • Advantage and disadvantage of BAL sequencing in comparison with biopsies need to be discussed.

Response:

When applied to clinical practice, it is expected that the diagnostic rate of lung cancer will be significantly improved as an advantage. On the other hand, disadvantages include prolonged examination time for bronchoscopy, increased invasion in patients, and development of complications associated with BAL (many of which are minor).

We added these descriptions to the Discussion section. Although there are few serious disadvantages, we would like to continue our study toward clinical application while paying careful attention to ethical aspects.

  • Prior art such as a report by Ryu (Pubmed ID: 31036768) for diagnosis of early-stage NSCLC have not been discussed.

Response: Ryu et al. reported the NGS analysis results of bronchial wash fluid obtained from 12 lung cancer patients, showing that the mutant allele fraction in bronchial washing fluid was approximately 4% on average. The mutant allele fraction in our BAL samples was an average of 11.2% and a median of 6%, and it can be said that the values were relatively high.

We cited this report in the Discussion section.

  • Figure 3 shows only the shared mutations between BAL and tumor. It is important to also show the discrepant mutations between these two sample types for each patient either in the same or a separate figure.

Response: Details of discrepant mutations are given in Supplementary Table 3. The etiology of the discrepant mutations cannot be explained scientifically, but seems an accidental phenomena based on the current data available; thus, it is deemed difficult to summarize them in a figure.

  • The detection sensitivity in blood samples appear lower than prior reports. How much was the starting volume? Could that explain lower than expected sensitivity?

Response: Approximately 10 ml of peripheral blood was drawn from the patients just prior to surgery, of which approximately 5-6 ml of plasma samples from each patient was used for analyses. The amount of DNA in plasma obtained from this amount of blood sampling is about 40 ng, which is considered to be sufficient for NGS analysis (required amount:> 1 ng).

We added these descriptions in the Methods section.

  • There seems to be discrepancies in tumor sizes between Tables 3 and S2. For example, the tumor sizes for patient 5 and 12 in Table S2 are 2.3 and 3.1 cm, respectively. But, in Table S2 those are listed as 0.7 cm.

Response: In Table 3, the size of invasive portion of the tumor was shown, not the tumor size itself. We are sorry for the confusing expression. Apart from that, one reviewer suggested the comparison between 30 and 2 cases is difficult to interpret, and therefore we decided to remove this table from the manuscript.

Thank you for your thoughtful comments.

Reviewer 3 Report

I would like to congratulate the Authors on their very interesting research. Indeed the use of BALB for diagnostic purposes is under investigated in the lung cancer field, although several papers are now providing evidence that BALB fluid is an excellent source of information in lung cancer management. So the article submitted by the Authors is both timely and very relevant for the lung cancer research field. Also, I praise the Authors for clearly stating the limitations of their study at the end of the discussion, and setting the ground for future studies. I am therefore very happy to recommend this manuscript for publication.

I have only two minor comments for the Authors:

  1. Figure 4: Probably due to an issue in the pdf conversion of the file, the venn diagram is not visible to me. This should be corrected.

  2. Methodology: I believe that lobectomy can cause some degree of contamination of the BALB fluid, due to the release of biological material (cells, blood, etc) during the surgery procedure. This potential caveat of the methodology adopted for the study should be mentioned in the discussion (e.g. in paragraph at lines 267-277 and then at the end of the discussion section together with the other study limitations). If, on the other hand, contamination cannot happen following surgery because of how lobectomy is performed, this should be fully explained to the reader for increasing the clarity of the manuscript.

Author Response

  1. Figure 4: Probably due to an issue in the pdf conversion of the file, the venn diagram is not visible to me. This should be corrected.

Response: Thank you for bringing it to our attention. All the reviewers pinpointed the deformity of this figure. We think it may be due to the computer glitch, probably caused by the different version of Word software we used (Word for MAC, version 16.58). We will ask the editor to solve this problem.

  1. Methodology: I believe that lobectomy can cause some degree of contamination of the BALB fluid, due to the release of biological material (cells, blood, etc) during the surgery procedure. This potential caveat of the methodology adopted for the study should be mentioned in the discussion (e.g. in paragraph at lines 267-277 and then at the end of the discussion section together with the other study limitations). If, on the other hand, contamination cannot happen following surgery because of how lobectomy is performed, this should be fully explained to the reader for increasing the clarity of the manuscript.

Response: Thank you for your good suggestion. One of the limitations of this study is that this ex-vivo BAL model uses surgically resected lung and that is not exactly the same as the BAL performed in clinical practice. In the case of surgically removed lungs, BAL may be contaminated with blood and bronchial epithelia during intraoperative manipulation, and under the situation where there is no blood flow, BAL recovery efficiency and cell-free DNA release from the tumor may be affected.

We added these descriptions to the Discussion section.

Thank you for your thoughtful comments.

Round 2

Reviewer 1 Report

The authors have satisfactorily addressed most of my concerns. 

Author Response

The authors have satisfactorily addressed most of my concerns. 

Response: Owing to your suggestions, our paper has been much improved. We also appreciate your high evaluation of our paper.

Reviewer 2 Report

Some of the responses to this reviewer’s questions are not very clear. For example, it is not clear where authors describe the benefits and disadvantages of biopsy. Also, in authors view, how is this work a significant advancement compared with the report by Ryu which used BAL from patients (not Ex Vivo) who had only early-stage tumors which were endoscopically invisible and achieved 95% concordance with tumor sequencing? In reference to Figure 3, this reviewer is not asking for the etiology of discordant mutations. All 53 mutations are already included in Figure 3 and therefore including discordant mutations will not require much additional space and will help the reader see the strength and limitations of the approach.

Author Response

Some of the responses to this reviewer’s questions are not very clear. For example, it is not clear where authors describe the benefits and disadvantages of biopsy.

Response:

In the case of bronchoscopic biopsy, lung cancer cannot be diagnosed unless the biopsy forceps are guided to the vicinity of the tumor under x-ray fluoroscopy and the tip of the forceps finally hits the lung tumor directly, so the actual correct diagnosis rate of lung cancer is around 30-80%, depending on the size and location of the tumor (doi: 10.1056 / NEJMoa1504601., doi: 10.18632 / oncotarget.18159.). In contrast, BAL can lavage the entire segment where the lung cancer lesion is located, so lung cancer can be easily diagnosed by routine procedures regardless of the size or location of the tumor. In addition, if this study can be translated into clinical practice and combination of BAL procedure and genomic analysis can be realized, the diagnosis rate of lung cancer is expected to be significantly improved.

On the other hand, when the BAL procedure is additionally performed with regular bronchoscopic biopsy, demerit such as prolonged bronchoscopy, increased patients’ invasion, and complications associated with BAL procedure may occur, although there seem few serious disadvantages regarding the BAL procedure.

We added these descriptions in the Discussion section.

Also, in authors view, how is this work a significant advancement compared with the report by Ryu which used BAL from patients (not Ex Vivo) who had only early-stage tumors which were endoscopically invisible and achieved 95% concordance with tumor sequencing?

Response:

For ethical considerations, we injected 50 ml x 3 times of physiological saline into segmental bronchus as an ex-vivo BAL model using resected lung, whereas in the study by Ryu et al., they performed bronchial washing (not BAL (bronchial alveolar lavage)) in vivo in lung cancer patients and injected 30 ml of physiological saline into segmental bronchus once.

Almost all lung cancer cases in Japan these days are endoscopically invisible tumors, and, therefore, only endoscopically invisible tumors were included in this study. In our study, we also included early-stage operable patients. The pathological stages of enrolled patients were IA in 16 cases, IB in 9 cases, IIA in 1 case, IIB in 2 cases, IIIA in 3 cases, and IVA in 1 case, and the distribution was biased towards earlier stages than those included in the study by Ryu et al. (IA in 1 case, IB in 5 cases, and IIB in 6 cases). On the other hand, regarding the diagnosis rate of lung cancer, in the study by Ryu et al., it was 10/12 (83.3%) because 2 out of 12 cases could not be diagnosed, whereas in our study, it was slightly better 30/32 (= 93.8%). However, considering that the patient background, methodology, and analysis method are slightly different, and the sample size is small in both studies, we think it is premature to judge that the results of our study with bronchoalveolar lavage were better. We believe what is important in this study is that the scientific validity of such a diagnostic approach has been demonstrated by the fact that the two independent research facilities showed similar results.

In this context, we added the descriptions in the Discussion section.

“The above-mentioned Ryu et al. performed bronchial washing in vivo in 12 patients with early-stage lung cancer and performed NGS analysis with resultant fluids. As a result, they reported encouraging results showing that concordance of driver mutations between bronchial washing fluids and primary tumors was 95.0%. This suggests that this type of diagnostic approach is feasible and promising, thus possibly opening a new avenue for “precision medicine” in lung cancer diagnosis.”

In reference to Figure 3, this reviewer is not asking for the etiology of discordant mutations. All 53 mutations are already included in Figure 3 and therefore including discordant mutations will not require much additional space and will help the reader see the strength and limitations of the approach.

Response:

As the reviewer suggested, the discordant mutations between FFPE and BAL fluid samples were summarized and added as Figure 4 in the manuscript.

Owing to your suggestions, our manuscript has been much improved. Thank you for your thoughtful comments.

Round 3

Reviewer 2 Report

Please examine image for the Venn Diagram (new Figure 5). Otherwise, no further concerns.